# Analysis of the Links between Social Intelligence and Coping Strategies of Business Managers in Terms of Development of Their Potential

**Lucia Zbihlejova** [1],[*] and **Zuzana Birknerova** [2]

1 Department of Intercultural Communication, Faculty of Management and Business, University of Prešov, 080 01 Prešov, Slovakia
2 Department of Managerial Psychology, Faculty of Management and Business, University of Prešov, 080 01 Prešov, Slovakia
* Correspondence: lucia.zbihlejova@unipo.sk

**Abstract:** The social as well as psychological development of individuals' potential is influenced by many factors, including managerial competences such as social intelligence and ways of coping with stress. This paper presents the links between social intelligence and strategies for coping with demanding situations by business managers, as well as gender differences in the perception of social intelligence and in the preference for coping strategies between male and female business managers. The research sample consisted of 149 business managers, of which 76 (51%) were male and 73 (49%) were female managers. The results were obtained through research based on two methodologies: MESI for the detection of social intelligence, and Brief COPE, designed to identify coping strategies. Based on the research results, it can be concluded that the links between the social intelligence factors and coping strategies, as well as gender differences in the perception of social intelligence and coping strategies by male and female business managers, have been confirmed. Research into the relationship between these two aspects and its results could contribute to the elimination of undesirable factors influencing the work process and to the subsequent development of the psychological, social and work potential of business managers.

**Keywords:** brief COPE; coping; MESI; social intelligence; stress





## 1. Introduction

Given the global situation of the ongoing COVID-19 pandemic, stress is a much discussed and current topic. The focus is on its harmful effects on human health, as well as a number of tips, techniques and efforts to cope with it and manage it with the least possible strain of mental energy. Social intelligence is a slightly lesser-known topic, but among experts it does not lag behind the issue of stress. It is the subject of research by many authors, both in terms of research and in terms of usability in practice, especially the application of knowledge in the field of leadership and work with people.

The presented paper deals with the issue of social intelligence, stress and its management and the connection between these two areas of psychology as one of the prerequisites for the development of potential in the field of business management. Research into the relationship between these two aspects and its results could contribute to the elimination of undesirable factors influencing the work process and to the subsequent development of the psychological, social and work potential of business managers.

## 2. Research Problem

The main objective of the presented research was to analyze the connections between social intelligence (as measured by MESI [1]) in relation to coping strategies (as measured by Brief COPE [2]) in the context of developing the potential of business managers.

Exploring gender differences is one of the essential and typical areas of research in the social sciences; therefore, we also aimed to detect the existence of gender differences in the preference of the studied phenomena.

## 3. Literature Review

People are social creatures; thus, they constantly find themselves in various social situations such as communication with people, quarrels, manipulation, assertion of their opinions, mutual cooperation, etc., which may be pleasant for the individual or, conversely, unpleasant. The following sections will, therefore, look into the theoretical background of concepts such as social intelligence, coping with stressful situations, and development of the potential of people, in this case, business managers.

### 3.1. Social Intelligence

Social intelligence is related to the definition of intelligence in general, so we encountered it as a concept only in 1920 [3]. However, it had been present as a phenomenon since the beginnings of human existence, when people first began to associate and socialize. It concerns the behavior of people in various interpersonal situations and tries to explain their actions and predict what is the basis for successful work with people, clients but also co-workers [4].

The conceptualization and operationalization of social intelligence (SI) are still debated in the literature. Discussions, which had developed as part of the conceptualization and the subsequent operationalization of social intelligence, attract the attention of authors to at least seven sets of issues [5]: (1) setting a nomological network which is agreed upon theoretically; (2) definition of the elements of the social intelligence construct; (3) pro-social versus anti-social nature of social intelligence in practical life; (4) personality versus psychometric approach to the measurement of social intelligence; (5) dispositional versus situational approach to studying social intelligence; (6) methodologies of studying social intelligence; (7) research on social intelligence within the cultural context. The definition of social intelligence vis-a-vis similar, related concepts (including emotional intelligence) is discussed herein.

The concept of social intelligence is not completely clear, as the distinction from other similar constructs (emotional intelligence, practical intelligence, academic intelligence, social skills, social influence, etc.) is ambiguous and some concepts overlap [6,7]. There are many definitions, and this is where the biggest problems arise. So far, scientists researching this area have only agreed that social intelligence is a multidimensional construct. However, there is no clear definition that would be recognized by all experts [8]. The definitions of social intelligence in the literature have certain features in common, namely the understanding of people, their inner world (thoughts and feelings) and, subsequently, an appropriate conduct in society. Social intelligence can therefore be used to describe the individual behavior of people in interpersonal situations, including those occurring in business and management.

Social intelligence is regarded in the managerial literature as a pivotal factor of managerial work regardless the position of the manager (top managers, middle managers, etc. [9]). Not only effectiveness of managerial work, but also the success of the whole organization depends on the ability of leaders and managers to use their social intelligence [10]. An effective leader or manager is often envisioned in the literature as socially and emotionally skilled and capable of self-management as well as managing others [11–13].

Ref. [14] claims that businesses investing in developing the social intelligence of their employees are more successful in the market than those which do not pay attention to them at all. On the basis of the results from nationwide studies, ref. [15] states that when hiring new workers, employers do not accentuate only their professional expertise, but focus also on such qualities which are connected to emotional and social intelligence. He also highlights that in the current labor market an emphasis is put on such competences as flexibility, teamwork, customer orientation, etc. These abilities require emotional and social intelligence aspects which keep playing a crucial role and become pillars of high performance in any occupation [15].



### 3.2. Coping with Stress

In recent years, stress has become a global, modern trend and an unwanted lifestyle for many people [16]. This is also the reason why many experts in the field of medicine, psychology, as well as management deal with it today. Stress occurs in various forms and shapes. In addition to negative experiences, stimuli, emotions and situations, stress can be caused by situations that are not negative, but we perceive them as stressful, such as promotion or expectation of the birth of a child. Based on this, it can be stated that the trigger of stress can be anything depending on the nature and personality of a particular person. For some people, it is enough to think about any aspect or about several factors that are gradually accumulating and causing tension [17].

Sources of stress can be various social situations, situations that directly threaten the life of the individual and various health problems, but also the people themselves, who consciously or unconsciously put pressure on the individual, for example in the form of exaggerated demands, humiliation, mistrust, physical or mental harm, etc. [18]. All these stimuli that burden or negatively affect a person and cause stress are called stressors [19,20]. According to [21], a stressor is a factor of the external environment that causes a state of stress in the body, i.e., a stress response. Appropriate stress response is a healthy and necessary part of human life. The key to success is not eliminating stress but managing it as it is responsible for physical relaxation [22]. Coping therefore represents any techniques and means that help the individual to cope with stress. Ref. [23] favor a similar formulation, perceiving coping as a set of cognitive and behavioral efforts aimed at managing, reducing or tolerating internal and external demands that threaten or exceed the resources of the individual.

Many strategies of coping with demanding situations and their classification occur in the literature [24–26]. For practical applications of stress coping strategies in the field of management, some authors have tried to create general strategies in terms of dispositional traits that managers could use in any stressful situations, although they do not fully capture the effects of everyday life [27]. On the other hand, each situation is examined as different; thus, each of them will require specific efforts and behavior [28].

Gender differences in coping strategies are the ways in which men and women differ in coping with psychological stress. Research that has studied the incidence of illness and its consequences on mental health suggests that individuals who use problem-focused coping show a lower risk of depression than those who use emotion-focused or problem-avoidant coping [29]. Ref. [30], for example, found that adolescent females were more likely than males to use ruminative coping and that this particular coping method was significantly associated with depression. Ref. [29] examined differences in coping and depression among Mexican Americans and found that problem avoidance was positively associated with depressive symptoms in both genders. In this context, ref. [31] adds that there is no gender difference in the effectiveness of managers, but there are gender differences in certain forms of their behavior.

### 3.3. Developing Potential of Business Managers

The first phase of the manager development process is the identification of their development potential. It defines the reserves of the manager, and thus creates space for the development and improvement of work results [32]. Based on a survey of requirements, ref. [33] lists the most important skills needed for business managers. These include problem solving and stress management, interpersonal skills including social intelligence, active listening, personal and career development, goal setting, and motivation. Managers' personal development plan sets out what they need to improve, what they need to develop in order to advance in their careers and streamline their work. According to [32], one of the possibilities is represented by coaching. Its elements are used by business managers to solve current problems, to develop their potential, to increase their performance, motivation, adapt to stressful situations and conflict resolution. According to [34], coaching focuses on increasing performance as well as developing the potential of business managers.

## 4. Methods

The research was carried out by means of a questionnaire, which consisted of two parts represented by two original methodologies: MESI methodology [1] for measuring social intelligence based on a psychometric approach and Brief COPE [2], which is designed to identify ways to cope with stress.

MESI contains 21 items assessed on a 5-point Likert scale, with a value of (0) representing the response "never" and a value of (4) representing "very often". These items saturate three factors of social intelligence: Empathy ($\alpha$ = 0.783), Manipulation ($\alpha$ = 0.854) and Social Irritability ($\alpha$ = 0.716).

The Brief COPE methodology was developed to assess the reactions of people who have to face a demanding or stressful event in their lives. It is an abbreviated version of the COPE methodology [35], which contains 60 items. This shortened version consists of 28 items, each 2 of them forming 1 of the 14 Brief COPE subscales (areas of coping strategies): Religion ($\alpha$ = 0.82), Substance use ($\alpha$ = 0.90), Active coping ($\alpha$ = 0.68), Planning ($\alpha$ = 0.73), Positive reframing ($\alpha$ = 0.64), Acceptance ($\alpha$ = 0.57), Humor ($\alpha$ = 0.73), Use of emotional support ($\alpha$ = 0.71), Use of instrumental support ($\alpha$ = 0.64), Self-distraction ($\alpha$ = 0.71), Denial ($\alpha$ = 0.54), Venting ($\alpha$ = 0.50), Behavioral disengagement ($\alpha$ = 0.65) and Self-blame ($\alpha$ = 0.69). The respondent responds to the items by choosing an answer from a 4-point Likert scale ranging from (1) "I haven't been doing this at all" to (4) "I've been doing this a lot". We determined the statistically significant correlations and differences using the statistical software SPSS 22.

## 5. Data Analysis

As stated above, the main objective of this study was to analyze the connections between coping strategies in relation to social intelligence in the context of developing the potential of business managers. We also aim to detect the existence of gender differences in the preference of the studied phenomena. Based on the stated research objective, we formulated three hypotheses:

**H1.** *There are statistically significant positive correlations between the three MESI factors of social intelligence of business managers and the individual Brief COPE coping strategies.*

**H2.** *There are statistically significant negative correlations between the three MESI factors of social intelligence of business managers and the individual Brief COPE coping strategies.*

**H3.** *There are statistically significant gender differences between the male and the female business managers in their perception of social intelligence.*

**H4.** *There are statistically significant gender differences between the male and the female business managers in their preference of coping strategies.*

The research sample consisted of 149 Slovak business managers, of which 76 (51.0%) were male and 73 (49.0%) were female managers aged from 26 to 58 years (average age of 37.5 years). The research was conducted in Slovakia, using the non-probability quota sampling method. Approached were all the businesses which are registered in the Business Register of the Ministry of Justice of the Slovak Republic and have their contact details freely available to the public. The research sample represents the number of the returned questionnaires (the participation was anonymous and completely voluntary). Business manager is identified here as someone who manages at least one other person and meets at least one of the following criteria:

- Active personal, telephone and electronic communication with clients for the purpose of strengthening and developing business relations.
- Care of current clients, active search and acquisition of new customers on the market, creation of a partner network.
- Commercial coverage of new products and services.

- Conducting business meetings, negotiating business terms and maintaining business standards.
- Creation, evaluation and optimization of business strategy in order to maximize the volume of sales and revenues, preparation and updating of sales plans.
- Market monitoring, marketing activities, reporting, preparation of market and sales analyses.
- Optimizing the portfolio of services offered in accordance with the development of the market and competition
- Participation in business meetings, seminars and presentations.
- Presentation of the portfolio of products and services, preparation of price calculations.

Business managers were selected both from the private (65%; $n = 97$) and the public sector (35%; $n = 52$) companies.

## 6. Results

We verified Hypotheses 1 and 2 using Pearson's correlation coefficient in SPSS22. The detected statistically significant correlations are presented in Table 1.

**Table 1.** Correlations between social intelligence factors (MESI) and coping strategies (Brief COPE).

| MESI<br>Brief COPE | Manipulation | Empathy | Social Irritability |
|---|---|---|---|
| Self-distraction | 0.156 | 0.104 | 0.100 |
| | 0.058 | 0.205 | 0.224 |
| Active coping | 0.089 | **0.312 \*\*** | −0.099 |
| | 0.282 | **0.000** | 0.231 |
| Denial | 0.056 | 0.118 | **0.308 \*\*** |
| | 0.496 | 0.152 | **0.000** |
| Substance use | **0.305 \*\*** | −0.036 | 0.114 |
| | **0.000** | 0.662 | 0.166 |
| Use of emotional support | 0.090 | 0.073 | 0.103 |
| | 0.273 | 0.375 | 0.211 |
| Use of instrumental support | **0.162 \*** | **0.240 \*\*** | 0.071 |
| | **0.049** | **0.003** | 0.387 |
| Behavioral disengagement | 0.128 | −0.056 | **0.369 \*\*** |
| | 0.119 | 0.500 | **0.000** |
| Venting | **0.311 \*\*** | **0.282 \*\*** | 0.141 |
| | **0.000** | **0.001** | 0.085 |
| Positive reframing | **0.252 \*\*** | 0.154 | −0.122 |
| | **0.002** | 0.061 | 0.140 |
| Planning | 0.020 | **0.226 \*\*** | −0.146 |
| | 0.810 | **0.006** | 0.076 |
| Humor | **0.346 \*\*** | **0.172 \*** | 0.071 |
| | **0.000** | **0.036** | 0.386 |
| Acceptance | **−0.161 \*** | 0.033 | −0.152 |
| | **0.050** | 0.692 | 0.065 |
| Religion | −0.074 | −0.003 | **−0.210 \*** |
| | 0.367 | 0.970 | **0.010** |
| Self-blame | **0.181 \*** | 0.146 | **0.200 \*** |
| | **0.027** | 0.076 | **0.015** |

\* Correlation is significant at the 0.05 level (2-tailed); \*\* Correlation is significant at the 0.01 level (2-tailed). Source: own elaboration in SPSS 22.

According to the data in Table 1, it can be stated that there are statistically significant positive correlations between individual factors of social intelligence and some stress coping strategies. Substance use and Humor were preferred by individuals who are prone to manipulative behavior. Manipulative people are also able to see the problem from its more positive side. Individuals prone to Empathy and Manipulation use Venting as a coping strategy, as these social intelligence factors correlate statistically significantly positively with the given coping strategy. Active coping, i.e., the direct management of stress using one's own strength, was chosen by individuals who excelled significantly in the Empathy factor.

There were also statistically significant negative correlations recorded between the Social Irritability factor and the Religion strategy. This means that socially irritated individuals do not seek to cope with stress through religious practices. Another statistically significant negative correlation occurred between the social intelligence factor of Manipula-

tion and the coping strategy of Acceptance. Manipulative business managers thus refuse to cope with stress actively and prefer other coping techniques—according to our results, these are Substance use, Use of instrumental support, Venting, Positive reframing, Humor and Self-Blame. The social intelligence factors did not statistically significantly correlate only with two coping strategies, i.e., Self-distraction and Use of emotional support, which means that business managers do not tend to seek or use these strategies to cope with the demanding situations they encounter.

Based on the above results, it can be stated that *H1 and H2* were *supported* and *confirmed*.

Hypothesis 3 (There are statistically significant gender differences between the male and the female business managers in their perception of social intelligence.) was assessed using a *t*-test for two independent selections in the statistical software SPSS22. The results of statistically significant differences are illustrated in Table 2.

**Table 2.** Gender differences in the perception of social intelligence of business managers.

| MESI Factor | Gender | M | SD | T | p |
|---|---|---|---|---|---|
| Manipulation | **male** | 1.7444 | 0.80688 | 4.519 | **0.000** |
| | female | 1.1937 | 0.67106 | | |
| Empathy | male | 2.4436 | 0.58255 | −0.186 | 0.853 |
| | female | 2.4599 | 0.47634 | | |
| Social Irritability | male | 1.4586 | 0.58662 | −0.136 | 0.892 |
| | female | 1.4716 | 0.57424 | | |

Source: own elaboration in SPSS 22.

According to the results shown in Table 2, it can be argued that in the social intelligence factor of Manipulation, male business managers score on average higher than the female managers. However, it should be noted that the results of the analyses are on a scale of disagreement. In other factors of social intelligence, we did not record statistically significant gender differences.

Based on this analysis, we can state that *H3* was *supported* in one of the three factors of social intelligence.

Hypothesis 4 (There are statistically significant gender differences between the male and the female business managers in their preference of coping strategies.) was assessed using a *t*-test for two independent selections in the statistical program SPSS22. The results of statistically significant differences are presented in Table 3.

According to the results shown in Table 3, it is clear that in the coping strategies of Denial, Use of emotional support, and Religion, female business managers scored higher. As a result, women are more likely than men to use other people's support during times of stress. Men scored more significantly in the Humor coping strategy. They are able to regard a situation with greater foresight and humor than women, who find it more difficult to detach themselves from the problem and therefore seek emotional support. Compared to the male managers, women also scored more significantly in the Religion strategy, which confirmed their stronger need to seek solace and support from someone else or in their faith.

**Table 3.** Gender differences in the preference for coping strategies between business managers.

| Brief COPE Strategy | Gender | M | SD | T | *p* |
|---|---|---|---|---|---|
| Denial | male | 1.2368 | 0.79780 | −1.995 | **0.048** |
| | **female** | 1.4658 | 0.59119 | | |
| Use of emotional support | male | 1.4342 | 0.69914 | −2.877 | **0.005** |
| | **female** | 1.7877 | 0.80328 | | |
| Humor | **male** | 1.9342 | 0.93122 | 2.982 | **0.003** |
| | female | 1.5068 | 0.81859 | | |
| Religion | male | 1.0658 | 0.95274 | −4.161 | **0.000** |
| | **female** | 1.7123 | 0.94985 | | |

Source: own elaboration in SPSS 22.

Regarding the individual coping strategies, statistically significant gender differences in their perception were found, although only in the factors of Denial, Use of emotional support, Religion (preferred by women), and the factor of Humor, which seems to be a more popular coping strategy among men. Based on the above, it can be stated that *H4* was *supported* in four coping strategies.

## 7. Discussion and Conclusions

The presented research confirmed existence of statistically significant correlations between the extracted social intelligence factors of the MESI methodology and some stress coping strategies of the Brief COPE methodology. In the addressed research sample, manipulative individuals were found to use positive stress coping strategies: Venting, Positive reframing, and Humor. This means that the business managers who can use others to their advantage have a positive mindset and can alleviate the stressful situation with humor or find something positive in it. Equally important, they can vent their negative emotions and not let these emotions suffocate them. However, these individuals have also developed a substance abuse strategy, which can serve as ventilation or a departure from reality, albeit in a negative sense.

Business managers, who are more strongly inclined to Empathy, prefer Active Coping strategies, i.e., they face the problem directly with the use of their own forces [36]. In addition, the Planning strategy was confirmed among them. Here, as in the previous strategy, the individual faces a problem and looks for solutions. In one case, based on a positive correlation coefficient, both empathetic and manipulative individuals agree in the choice of the coping strategy of Venting. This suggests that people who feel good in society, are able to understand others or use them, can give their negative emotions free passage. Respondents who scored more significantly in the Social Irritability factor show a tendency to choose Behavioral disengagement, Denial, and Self-blame as a way of coping with the burden. These individuals refuse to admit reality and problem and very quickly give up their efforts to solve it. In addition, Social Irritability correlated negatively with the Religion strategy, which means that socially irritated business mangers do not seek to cope with stress through religious practices.

A similar study [11] aimed at the analysis of coping strategies and social intelligence showed some identical results, although using other methodologies: TSIS [37] and SVF 78 [38]. Consistency was found in some strategies for the factor of Empathy, which corresponds to the factor of the TSIS methodology Social information processing, specifically in the strategies: Diversion, Control of the situation and reactions, and Positive self-instruction. Similar results were found in another TSIS factor which, together with the previous one, correlates highly positively with the factor of Manipulation [1]. According to this research, socially skilled individuals have Escape tendencies which, to some extent, coincides with the use of addictive substances by manipulators in the presented research. Another similarity was confirmed in the relationship between Social Irritability and the Resignation strategy, as a negative Pearson correlation coefficient was found in

the Social Irritability factor with all three SI factors of the TSIS methodology and the Resignation strategy was negatively correlated with all social intelligence factors of the TSIS methodology [1].

Both studies found significant gender differences in the assessment of the factors examined. In the need for emotional support coping strategy, studies clearly agreed that this strategy is preferred by women, who generally find it more difficult to cope with stress and seek advice and help when talking to loved ones [11]. The second significant gender difference is confirmed by both works in the underestimation of the stressful situation by men. Men are able to break free from stress more easily and help each other by trying not to attach great importance to a difficult situation. Thus, the presented studies confirmed the existence of statistically significant correlations in some coping strategies and social intelligence factors. However, some diversity was also observed, probably due to differences in the number of respondents, in the fact that the other research sample consisted of employees of one particular company while this research sample consisted of business managers from various areas of management, business and trade, and also due to using different methodologies and other aspects.

Ref. [39] conducted research on gender differences in the perception of stress and its management, but on a much larger sample of respondents (2816 respondents, of which 1566 were women and 1250 were men). In this research, women scored significantly higher than men on emotional and avoidant coping styles and lower on rational coping styles and coping through dissociation from the problem. Men have been found to have more emotional inhibitions than women, with women experiencing more stress than men and their stress patterns being more emotion-focused than men.

It can be noted that gender differences in the area of business and trade were also part of the study by [40], who found statistically significant gender differences in the perception of personal selling from the customer's perspective. Another study by [41] was evaluated consumer shopping behavior in the context of gender equality. Its results confirmed the existence of statistically significant differences between male and female consumers when evaluating shopping behavior.

Based on the obtained results, it can be stated that coping with stress can to some extent affect the ability of an individual to respond in social situations. Conscious or unconscious choice of a specific coping strategy can cause uncertainty in a person's work and thus create a barrier in the development of his mental and work potential. It is therefore appropriate to research the issue to a greater extent and with new knowledge to encourage the development of techniques that could alleviate or eliminate potential problems and thus contribute to improving human personal and psychological potential in the work process as well as personal life of individuals.

In terms of exploring social intelligence and coping with stress in a cultural context, it is possible to define at least two sets of issues. The first relates to solving the level of universality of the extracted factor structures of these constructs within individual cultures. The second is connected to the assessment of the individual specified factors of these phenomena within different cultures. This research was limited to the sample of Slovak business managers, which represents a possible starting point for future research studies that could take into account the cultural dimension of the issue at hand.

**Author Contributions:** Conceptualization, L.Z. and Z.B.; methodology, L.Z.; software, Z.B.; validation, Z.B.; formal analysis, Z.B.; investigation, L.Z.; resources, L.Z.; data curation, Z.B.; writing—original draft preparation, L.Z.; writing—review and editing, L.Z.; visualization, L.Z.; supervision, Z.B.; project administration, L.Z.; funding acquisition, Z.B. All authors have read and agreed to the published version of the manuscript.

**Funding:** This research was funded by the grant project KEGA 012PU-4/2020 (Trading Behavior—Creation of the subject and textbook for non-economic study programs).

**Institutional Review Board Statement:** Not applicable.

**Informed Consent Statement:** Informed consent was obtained from all subjects involved in the study.

**Data Availability Statement:** The data presented in this study are available on request from the corresponding author. The data are not publicly available due to the privacy and ethical restrictions.

**Conflicts of Interest:** The authors declare no conflict of interest. The funders had no role in the design of the study; in the collection, analyses, or interpretation of data; in the writing of the manuscript; or in the decision to publish the results.

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
