# Peer review of "Analysis of the Links between Social Intelligence and Coping Strategies of Business Managers in Terms of Development of Their Potential"

_societies, doi:10.3390/soc12060177_

Round 1

Reviewer 1 Report

Dear writer(s). Your topic is promising although its contribution to the field is limited. In its current stage your manuscript (and maybe research) needs major revision to meet the publication standards of Societies.

Please see below the areas that need to be improved.

I see major problems in the research design and the lack of information concerning your methods:

-          Where did you conduct the research (which country)?

-          How did you select your samples (sampling method)?

-          From which sectors are the businesses your participants work for?

-          Are the participating managers working for SMEs, large or multinational companies?

-          What defines a manager?

-          You do not share the questionnaires and therefore the classifications you use cannot be verified.

- You have no section on informed consent and ethics approval.

Your research focusses on male/female participants, while a focus on different personality types (manager types) would have been more helpful. It could be that women in managing position represent a specific personality type which helped them to reach their leadership role in a more competitive male dominated environment. The country in which the research took place is as well important, as the cultural impact has to be considered. Female managers from Denmark will show different responses than female mangers from China. The cultural aspect of a society is an important differentiator that your research currently lacks.

The literature you refer to is very limited and is dominated by eastern European authors. You need to provide more and a more diverse critical review of the literature.

In your discussion and conclusion you mention “…these individuals have also developed a substance abuse strategy…”. With the provided data I cannot see any prove for this conclusion. Your participants must have stated in the questionnaires that they have developed a substance abuse strategy for you to draw such a conclusion. Again, the questionnaires have not been provided as support.

Your manuscript needs major spelling and editing support (see as well line 215 to219).

Reviewer 2 Report

I have no objections to the practical part of the article.

In my opinion, the theoretical part should definitely be extended in all three subsections, the text should be twice as large. The number of the cited literature should also be increased to at least 40 items.

In the methodological part, there is no information in which country (s) the research was conducted and at what date.

Author Response

The comments have been considered and implemented within the paper revision.

Round 2

Reviewer 1 Report

Dear author(s),

Your manuscript still lacks to explain how decided who will participate in your research and you distributed the surveys.  "Non-probability quota sampling method" needs further information to ensure non-biased and representative sampling. "Business managers were selected both from the private (65%; N = 97) and the public 203 sector (35%; N = 52) companies" needs information how you decided which company should participate.

The linkt to the MESI factor questionaire leads to another research from 2013. I am not sure that this is the right link as the questionnaire for this research had been requested.

Otherwise the revision covered all comments.

Author Response

Dear Reviewer,

thank you for the further comments. We will try to respond to both points:

1) The data procedure - we approached all the businesses which are registered in the Business Register of the Ministry of Justice of the Slovak Republic and have their contact details freely available to the public. Thus, we approached all of these and the research sample in the article represents the number of the returned questionnaires (we do not know the exact number of the addressed individuals as it was all up to the company and the participation was anonymous and completely voluntary). 

2) The link to MESI represents a link to the full text of the published paper where the methodology is content-specified (factor analysis completely included). We enclose the questionnaire which was distributed to the respondents (together with Brief COPE and with a few demographic items as specified in the research sample).
